# Spatial Configuration Effects on the Dissociation between Active and Latent States in Visual Working Memory

**DOI:** 10.3390/bs13080636

**Published:** 2023-07-31

**Authors:** Ziyuan Li, Qiang Liu

**Affiliations:** 1Research Center of Brain and Cognitive Neuroscience, Liaoning Normal University, Dalian 116029, China; 15097977739@163.com; 2Institute of Brain and Psychological Sciences, Sichuan Normal University, Chengdu 610000, China

**Keywords:** visual working memory, active state, latent state, dissociation account, spatial configuration

## Abstract

Visual working memory includes both active, recallable items and latent items not directly available for recall. During the online manipulation of active working memory, latent items gain robust retention. According to the dissociation account, active and passive memories exist in independent states, both of which take up their own cognitive resources. However, it is unclear whether dissociation is a universal phenomenon during memory maintenance. Given that memory information is retained as a binding of identity and location, the spatial proximity of memory items might shield the dissociation property. To test this, we adopted a retro-cue memory task where the cued and uncued items were separated in the active and latent states. In Experiment 1, the memory items were presented at a relatively large spatial distance. The results supported the dissociation account for well-separated items. However, Experiment 2 rejected the dissociation for closer-in items, possibly because items in visual working memory were spatially labeled. These findings suggest that while memory maintenance generally conforms to the dissociation account, the spatial configuration of memory items can affect the dissociation property between the active and latent neural states in visual working memory.

## 1. Introduction

Visual working memory (VWM) plays a crucial role in various cognitive processes and our day-to-day functioning. It is responsible for temporary memory storage and manipulation that allows us to actively hold and process information, even in the absence of immediate sensory input [1,2]. WM systems play a vital role in various high-level cognitive functions, such as problem solving and reasoning [3,4,5,6]. By understanding the key role of VWM in cognitive processing, we can appreciate its significance in various domains and its impact on our ability to perform complex tasks effectively.

For many decades, the dominant theory of working memory was that VWM information primarily depended on persistent neural activity [7,8]. It was thus often assumed that the continued retention of mental memory necessitated persistent neural firing activity. However, as more work has aimed to explore VWM more deeply, the traditional view that working memory information is retained in persistent activity has been challenged [9,10,11,12]. For example, in a delayed memory task, a ramp-up of neural activity from baseline was observed during the late retention period [13,14]. These results suggest that the memory information could be retained in the mind in a silent pattern. The synaptic theory of working memory proposed that working memory information could be maintained in an activity-silent manner via short-term synaptic plasticity [14,15,16,17,18,19]. In contrast to the active state accompanying persistent activity, the “activity-silent” format was referred to as latent state (i.e., passive state, silent state, activity-silent state) in the current study for consistency.

Though the latent memories failed to be detected by standard recording techniques, multiple pieces of evidence reliably supported the genuine retention of memory information in the latent state. For example, following the administration of visual impulse stimulation or transcranial magnetic stimulation (TMS) to targeted brain cortex, latent memories can be reactivated after their neural activity drops to baseline [16,17,18]. In terms of energy consumption, persistent firing underlying active memory retention is assumed to be energetically expensive, while short-term synaptic plasticity patterns of memory retention have a clear cost advantage with more efficiency [10,14]. Thus, the latent state maintenance, in combination with the active state, likely manifested an evolutionary advantage. From a functional perspective, currently task-relevant information was held in the active state, which was responsible for immediate online processing, while the latent state provided robust maintenance for prospectively relevant items [20,21]. The VWM items can flexibly transition between the active and latent states according to the task context [14,22,23].

The state-based models demonstrated that working memory was dynamic, evolving in a task-dependent way [11,24]. With respect to VWM maintenance with dual states, Li et al. (2020) proposed a resources-dissociation account. This account proposed that the resources in the active and latent states were mutually independent [25,26]. Specifically, the memories in the latent state were unaffected by the variation in active memory load, and the reverse was true. Based on the independence between the two distinct states, the cognitive processing in each storage state was endowed with relatively high efficiency during multi-state maintenance. Though memory information suffered loss due to the representational state switch [27], the dissociation property of two states manifested a positive contribution, which ensured that the latent memories remained unaffected by “irrelevant” active processing, thereby avoiding memory loss in general.

However, it was unclear whether the dissociation was a universal phenomenon during the multi-state maintenance. With regard to the memory stimuli from the visual world, the obligatory encoding of location was applied to working memory processing [28,29]. Previous studies have suggested that locations are obligatorily encoded along with features, even if the locations are task-irrelevant [30,31]. Indeed, in a location change detection task, participants were required to treat the objects as placeholders. Researchers found that location memory in the picture plane was disrupted when the color or shape of an object changed. This effect was similar for both real-world objects and rectangles with similar aspect ratios [32]. This research supported the proposal that working memory items were bound to the global spatial configuration of memory stimuli. In addition, when items received attention, there was a spontaneous integration of features and location in VWM. It has been shown that bindings of location and feature showed high vulnerability to location changes, suggesting that location played a vital role in binding retrieval and their initial encoding [29,32]. According to Gestalt principles, memory items located in close proximity tended to be chunked as “one object” [33]. Furthermore, the residual trace of memory items might potentially introduce interaction or interference across memory items due to their original spatial “adhesion”, though those task-irrelevant items were offloaded into the latent state. Thus, it was hypothesized that the spatial arrangement of memory items would disturb the dissociation of active and latent states during dual-state maintenance.

This study aimed to test whether the spatial configuration of memory items had an effect on the dissociation property between active and latent memories during visual WM maintenance. In this study, we adopted a retro-cue memory task where the cued items were held in the active state and the uncued items were offloaded to the latent state [25,34]. The active memory load was modulated in the two behavioral experiments. In Experiment 1, memory items were presented at a relatively large spatial distance. We expected to replicate the dissociation account by observing no active load effect on latent memory maintenance. Whereas in Experiment 2, the spatial gap between some memory items narrowed. If the spatial configuration of memory items impacted the dissociation property, latent memory maintenance would be impacted by active load variation, resulting in the absence of independence between active and latent memory states.

## 2. Experiment 1

### 2.1. Materials and Methods

#### 2.1.1. Participants

The key measure was the active load effect on memory performance in distinct states. We used G-power to calculate the appropriate sample size, which yielded a minimum of 19 to achieve the effect size (0.8), with 80% power and 0.05 α error. Chinese college students from various academic majors were recruited to participate in our study. A sample of 23 participants (4 males; age: 21.69 ± 2.71 years) was collected in Experiment 1. All of them signed informed consent when arriving at the laboratory. Each participant had no mental illness, and the vision or correct-to-vision was normal. Considering color as a memory stimulus, they reported no color blindness or color weakness.

#### 2.1.2. Stimuli and Procedure

E-prime 2.0 was used to run the experimental procedure. The size of the LCD monitor was 19 inches with a 60-Hz refresh rate and 1920 × 1080 pixels. Memory stimuli consisted of colored squares with a size of 0.49° × 0.49°. Throughout the trial, a black fixation cross with a size of 0.23° remained visible at the center of the monitor screen, serving as a constant reminder for participants to concentrate on each trial. The memory response is recorded by pressing the keys. The stimuli’s colors were selected from a color pool that consisted of easily distinguishable colors (blue, black, red, magenta, green, lime, cyan, white, yellow, and purple). There was no repetition of memory color within a trial. In Experiment 1, all color squares in two memory arrays were distributed on an invisible approximate square (2.62° × 1.8°) around the center fixation cross against the gray background (see Figure 1). Four items in memory array 1 (M1) occupied the four corners, and items in memory array 2 (M2) were located above or below the fixation cross. Participants’ viewing distance from the center of the monitor was 70 cm.

The procedure schematic is depicted in Figure 1. A blank retention with fixation cross lasted for 0.5 s to initiate the memory task (not shown in the schematic). Furthermore, the presence of M1 lasted for 0.5 s. After a 1-second delay, a retro-cue appeared for 0.2 s, which randomly indicated upward, downward, left, or right. For example, when the first retro-cue pointed in the upward direction, one of the upper two items was possibly detected in Probe 1. Furthermore, M2 was presented after the first retro-cue offset and a 1.5-second delay. The items in M2 were presented along the midline. Importantly, two manipulations were applied to M2. In 50% of trials, M2 presented two items, with one presented above and the other below the fixation cross. In the other 50% of trials, only one item was presented either above or below the fixation cross. Following a delay of 1 s, Probe 1 appeared. Participants had to determine whether the item in Probe 1 matched the color of the initially cued items in M1 and items in M2. After responding to Probe 1 and a delay of 0.5 s, the second retro-cue appeared, indicating which of the four items in M1 would be tested in Probe 2. Similar instructions are given for Probe 2, where participants need to identify whether the probe item has a different color from the color of items cued by second retro-cue. If the probe items have changed, then press “M”, or press “Z”. There was a 50% probability of making a “Z” response for the two probes within each trial. The probe arrays were always visible unless they made a response. In this task, accuracy was stressed over speed.

Within a trial, the storage state of memory items was separated following the first retro-cue. The cued items in M1 and items in M2 were held in an active state because these items would be detected immediately in Probe 1. Whereas uncued items in M1 transitioned to the latent state. For the active memories, the load ranged from three to four, and the latent memory load did not change (two items). Notably, in order to verify the items in M1 were indeed held in different states after the first retro-cue, the second retro-cue would randomly point to one of the four items in M1, which thus resulted in two trial types: (1) memory item was pointed by retro-cues two times, thereby constantly held in the active state; (2) memory item was pointed by the second retro-cue only, thereby kept in the latent state following the first retro-cue onset. In type 1 trials, Probe 2 was the indication of active memory performance, while Probe 2 reflected latent memory performance in type 2 trials.

Before the formal trials, participants should perform at least 12 trials for practice in each experiment. They would receive the feedback only during the practice trials. Each experiment included six blocks, resulting in 192 trials in total. To relieve fatigue, a pause of at least 30 s between blocks was provided. Moreover, we informed participants not to involve verbal encoding [35].

### 2.2. Data Analysis

Memory accuracy was the primary measure for data analysis. Firstly, we explored whether the load of M2 had a distinct effect on the first cued items in M1 and the items M2. Thus, the analysis was conducted by a 2 × 2 repeated measures ANOVA with factors of load and memory array type for Probe 1. Importantly, our interest was to assess whether active memory load had an effect on memory in the latent state. Based on the current experimental procedure, active memory performance was also indicated by Probe 1 and type 1 of Probe 2. Considering that the probe item in type 2 of Probe 2 will likely be detected again following Probe 1, we calculated the memory accuracy of Probe 1 as active memory performance. Meanwhile, the accuracy data from type 2 of Probe 2 was selected as latent memory performance. The memory accuracy was analyzed by repeated measures ANOVA with factors of storage neural state and active memory load. A post-simple effect analysis would be conducted following a significant interaction. We used JASP 0.16.4 software to perform the data analysis [36].

### 2.3. Results

As plotted in Figure 2 (left), the results from a 2 (active load 3 vs. active load 4) × 2 (memory array 1 vs. memory array 2) repeated measures ANOVA showed a significant main effect of load, F (1, 22) = 21.281, *p* < 0.001, η^2^*p* = 0.492. The effect of memory type was also significant: F (1, 22) = 9.135, *p* = 0.006, η^2^*p* = 0.492. However, the interaction between load and memory type was not significant, F (1, 22) < 1. The accuracy of the two arrays in the low load condition was higher than that in the high load condition. The accuracy was significantly better in memory array 2 than in memory array 1. In such a sequential memory task, two memory arrays were presented sequentially, so they strategically perceived the cued items in memory array 1 and the items in memory array 2 as two events, though these items were held in the same state. Memory array 2 produced better performance because items in this array were probed immediately.

Moreover, a 2 (active state vs. latent state) × 2 (active load 3 vs. active load 4) repeated measure ANOVA was conducted (Figure 2 right). A significant main effect of state was observed (F (1, 22) = 8.534, *p* = 0.008, η^2^*p* = 0.279), while the main effect of active load did not achieve significance (F (1, 22) = 4.440, *p* = 0.514, η^2^*p* = 0.020). Importantly, there was a significant interaction between the two factors: F (1, 22) = 5.870, *p* = 0.024, η^2^*p* = 0.211. A paired-sample *T*-test analysis was then performed. The results suggested that the accuracy of active memories significantly decreased as the active load increased (t (22) = 3.451, *p* = 0.002, Cohen’s d = 0.719). Whereas the accuracy of latent memories did not differ between the two active load conditions (t (22) = 0.863, *p* = 0.397, Cohen’s d = 0.180). These results suggested that increasing the memory load of the active state significantly impaired the active memories only, and exerted no effect on the latent memories.

That replicated the previous study in a slightly modified retro-cue paradigm, conforming to the dissociation account between active and latent states. These results paved the way for further exploration. If the spatial configuration of memory items plays a crucial role in determining the interaction between active and latent states, presenting memory items in close proximity would cover the dissociation property between active and latent states. The following experiment attempted to explore this hypothesis.

## 3. Experiment 2

### 3.1. Materials and Methods

#### 3.1.1. Participants

Another 21 participants (6 males; age: 22.24 ± 2.16) were recruited in this experiment who never took part in Experiment 1. They were required to sign an informed consent form before the start of the trials. Each participant reported normal vision or corrected-to-normal vision, as well as normal color vision.

#### 3.1.2. Stimuli and Procedure

The stimuli and procedure in this part were identical to those used in Experiment 1, with the following modification: The spatial arrangement of memory items was modified. The items in M1 and M2 were all located on an invisible rectangle (2.62° × 1.3°) around the fixation cross. That is, there was a shorter vertical gap between items in both arrays (see Figure 3). Participants were provided with the same instructions and requirements as in Experiment 1 to perform this experiment.

### 3.2. Data Analysis

A similar analysis was conducted in this part. Firstly, we explored whether the load of memory array 2 had a distinct effect on the first cued items in M1 and the items in M2. Thus, the analysis was conducted by a 2 × 2 repeated measures ANOVA with factors of load and memory array type for Probe 1. The active memory performance is indicated by Probe 1, while the accuracy data from type 2 of Probe 2 was selected as latent memory performance. Furthermore, the memory accuracy was analyzed by repeated measures ANOVA with factors of storage neural state and active memory load. A post-simple effect analysis would be conducted following a significant interaction. We used JASP 0.16.4 software to perform the data analysis [36].

### 3.3. Results

First, the results from a 2 (active load 3 vs. active load 4) × 2 (memory array 1 vs. memory array 2) repeated measure ANOVA showed a significant main effect of load: F (1, 20) = 7.119, *p* = 0.015, η^2^*p* = 0.262. The effect of memory type was not significant: F (1, 20) = 0.065, *p* = 0.802, η^2^*p* = 0.003. There is also no interaction between load and memory type: F (1, 20) = 0.030, *p* = 0.863, η^2^*p* = 0.002. These results were depicted in Figure 4 (left), which showed that the modulation of active load was effective and exerted an influence on the two arrays. Different from Experiment 1, the memory array 2 and the cued items of memory array 1 were conceived as “one object”, though they were presented sequentially. Thus, we presumed that the spatial context might have an influence on memory encoding and retention processing.

We administered a 2 × 2 repeated measure ANOVA (Figure 4 right). An interaction between the storage state and active load failed to reach significance (F (1, 20) = 0.821, *p* = 0.376, η^2^*p* = 0.039), while we observed a significant main effect of state (F (1, 20) = 9.237, *p* = 0.006, η^2^*p* = 0.316), and active load (F (1, 20) = 10.764, *p* = 0.004, η^2^*p* = 0.350). These results suggested that the memory items were successfully separated into two distinct states. The uncued items were held in the latent state, and the cued items and memory array 2 were kept in the active state. Importantly, active load had an effect on both the active and latent memories. That yielded the conclusion that, when memory items are presented in close proximity, the dissociation account does not hold true.

## 4. Discussion

The primary goal of this study was to assess the role of the spatial configuration of memory items in the dissociation account between the active and latent states. To this end, a retro-cue paradigm was adopted to separate the storage states of memory representations. The active load ranged from three to four, and the latent state had a fixed load. The absence of active load variation having an effect on latent memories indicated the dissociation between the two states. In Experiment 1, memory items were located on an invisible square with a relatively large gap between them. The results suggested that the variation of active memory load affected latent memory performance, consistent with the dissociation account. In Experiment 2, memory items were in close proximity, and the results showed that latent memories were greatly impaired when increasing the active memory load. These results indicated that the dissociation property did not hold true when the spatial configuration of memory items changed. The current findings concluded that, while the active and latent states generally conform to the dissociation account, the spatial configuration of memory items had an effect on the dissociation property in VWM.

While previous research has supported the idea that active processing does not affect latent memory maintenance, the current study demonstrated for the first time that when memory items are presented in close proximity, active memory processing can indeed influence latent memory maintenance. That revealed that the dissociation between active and latent states is not a universal phenomenon but is contingent upon the spatial context in which memory items are presented. As mentioned in previous studies, spatial location indeed plays an important role in VWM [29,31,32,37]. With respect to the multiple mechanisms of representational maintenance, it was reasonable to observe the phenomenon that retaining a latent set along with an active set yields overall better performance, either high accuracy or faster reaction time, than holding two active sets concurrently [20,26,38]. The current study provided a new avenue for achieving perfect independence of active and latent states, generating little memory loss in terms of overall memory performance.

According to the current findings, there are some possible explanations for accounting for the influence of spatial proximity on the interaction between active and latent memory maintenance. From the perspective of interference and competition, when memory items are in close spatial proximity, the neural representations of the active memory might overlap or interfere with the neural representations of the latent items. Alternatively, it could be assumed that, if items are tightly packed together, the neural resources allocated for active memory maintenance may spill over or encroach upon the resources required for latent memory storage. Either the inference account or the resource allocation account can reasonably interpret a degradation or impairment of the latent memory storage. In addition, in view of binding, when memory items are in close proximity, there may be a greater tendency for binding between nearby items, making it challenging to maintain a clear separation between the active and latent states. That might be the underlying reason for the observed absence of independence between active and latent states. It was necessary to explore the potential mechanisms through further research.

These explanations mentioned above focused on the interference of active memory maintenance with latent memory retention. However, the degradation of latent memory performance might result from the suppression of latent memory items. More specifically, spatial proximity can influence attentional processes involved in memory maintenance. When some items are close to each other within an array, attention resources may be biased toward the active memory items, which are immediately task-relevant, leading to the suppression of latent memory items. This attentional bias could potentially impact the encoding, consolidation, and retrieval of latent memory representations [39,40]. The current finding provided no evidence to distinguish these possible mechanisms. Therefore, it is necessary to conduct additional research using neuroimaging techniques, computational modeling techniques, or EEG techniques to provide more insights into the mechanisms by which spatial proximity influences the interaction between active and latent memory maintenance.

Indeed, Oberauer (2002) administered a memory updating task where participants performed arithmetic operations on the digits within the active set, and they found that the setsize effect of active and latent sets did not interact [26]. Consistently, Li et al. (2021) have found no effect of active load on latent memories in both retro-cue memory tasks and sequential presentation memory tasks [25]. Different from digit stimuli in Oberauer’s research, Li et al. chose simple color as memory stimuli, which effectively prevented the contamination of verbal encoding. The current study provided further support for the resource-dissociation account and shed light on its reliability; on the other hand, it indicated the limit of dissociation in some specific spatial contexts within the state-based theory of visual working memory.

The current study provided an appropriate paradigm for exploring VWM maintenance with dual-state. After the presence of a retro-cue, the cued items received more resources, representing a higher priority in the active state, while the uncued items lost attentional resources, thereby offloading them to the latent state with a lower priority. Thus, the priority of memory items induced the separation of storage neural states following the presence of retro-cues during memory information retention [23,41]. In addition to the retro-cue memory task, a sequential presentation memory task can also direct memory items into distinct storage states. In that memory task, the first presented memory array transitioned into the latent state during the online processing of the second memory array. The separation of storage states was based on the relevance of memory items to the immediate task [22,25,27].

Understanding the influence of spatial configuration on the dissociation of active and latent states has practical significance in various domains outside of the context of research. For example, in designing working memory tasks or training programs, it may be important to consider the spatial arrangement of memory items, such as ensuring they are not too closely located, to optimize memory performance. In addition, there is valuable instruction for creating optimal learning environments. By organizing learning materials or information in a way that minimizes interference between currently available information (active items) and prospectively relevant information (latent items), educators can improve students’ retention and retrieval of information. In addition, within the field of software development or human-computer interaction, knowledge of the spatial arrangement effect can guide the placement and organization of elements to minimize cognitive load and enhance user performance. Developers can create interfaces that optimize information processing and facilitate efficient task completion.

Nevertheless, this study does have potential limits. In the current retro-cue memory paradigm, the latent items were initially held in the active state together with active memory items before the indication of the first retro-cue. It was unclear whether the effect of spatial configuration was still observed when active and latent memory items were sequentially presented but overlapped in location. In addition, it was necessary to examine whether the effect of spatial arrangement was applicable when shape or orientation acted as memory stimuli in further study. Further research is needed to explore the specific factors that modulate the dissociation property between active and latent states. Moreover, although efforts were made to recruit an adequate number of participants, the overall sample size may be considered relatively small. While these numbers were sufficient for the scope of our study, a larger sample size would allow for more robust statistical analyses and potentially enhance the generalizability of the results. Future studies with larger and more diverse samples would be valuable in corroborating and extending our findings.

## 5. Conclusions

During the dual-state maintenance of VWM, active and latent memories could be independent of each other in some scenarios. However, the present study highlights the significant role of spatial arrangement in determining the interaction between the two states. Specifically, when memory items are closely located, the resources allocated to active processing can spill over and have a negative impact on the maintenance of latent memories. These findings demonstrate that the dissociation between the active and latent states is not a fixed property but can be influenced by the spatial configuration of memory items. It is therefore crucial to consider the spatial arrangement of memory items to optimize WM performance.

## Figures and Tables

**Figure 1 behavsci-13-00636-f001:**
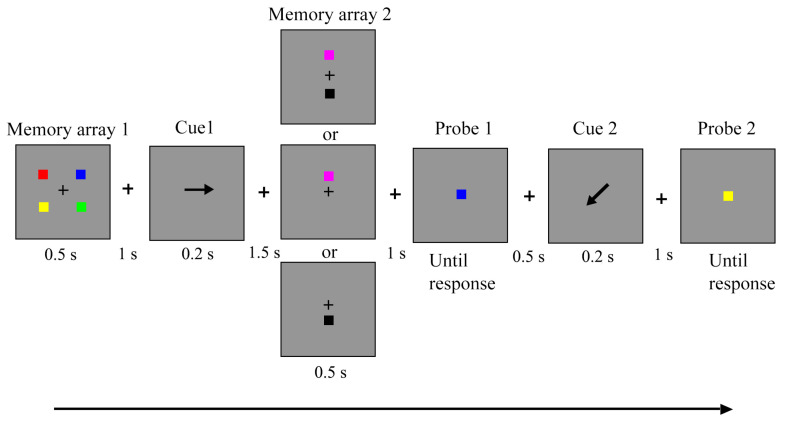
The schematic of trial events in Experiment 1 shows the values indicated for the time span.

**Figure 2 behavsci-13-00636-f002:**
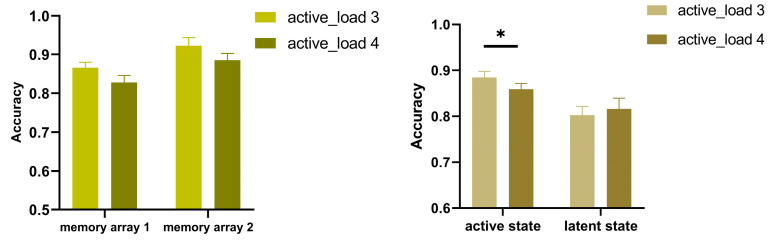
Behavioral accuracy in Experiment 1. The mean accuracy in the two memory arrays under the two load conditions (**left**). The mean accuracy in the active and latent states under the two load conditions (**right**). Error bars denote SEM. * *p* < 0.05.

**Figure 3 behavsci-13-00636-f003:**
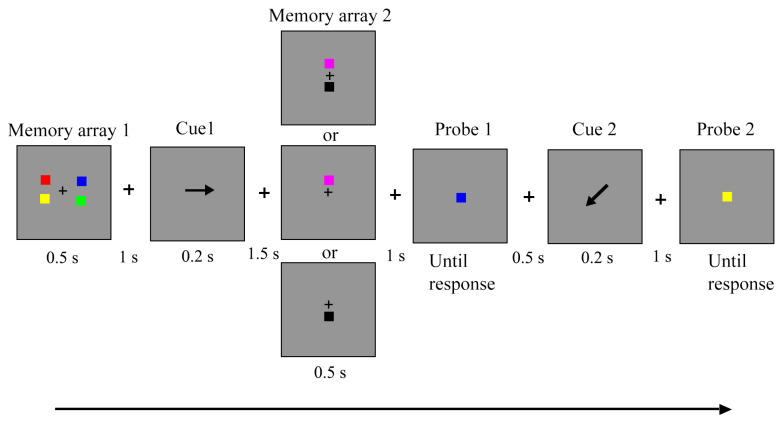
The schematic of trial events in Experiment 2. The values indicated the time span.

**Figure 4 behavsci-13-00636-f004:**
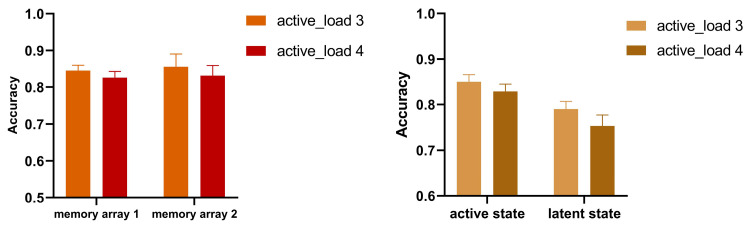
Behavioral accuracy in Experiment 2. The mean accuracy in the two memory arrays under the two load conditions (**left**). The mean accuracy in the active and latent states under the two load conditions (**right**). Error bars denote SEM.

## Data Availability

The data that supports the findings of this study is available in the repository (https://osf.io/us4n9/). Access on 18 June 2023.

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
