# Peer review of "Spatial Configuration Effects on the Dissociation between Active and Latent States in Visual Working Memory"

_behavsci, 2023, doi:10.3390/bs13080636_

Round 1
Reviewer 1 Report
This study investigated the Spatial configuration effects on the dissociation between active and latent states in visual working memory. The study is very interesting.
Nevertheless, I have some comments and concerns, listed below:
Introduction:
- Lines 28-31: “Visual working memory (WM) plays a crucial role in various cognitive processes and our day-to-day functioning. It is responsible for temporary memory storage and manipulation that allows us to actively hold and process information, even in the absence of immediate sensory input.”
General references should be noted (i.e., Baddeley) concerning working memory for these sentences.
Experiment 1 and experiment 2:
Concerning participants, information on the type of study would be interesting. Are the participants all doing the same studies? The authors should provide details.
- Lines 114-115: “A sample of 23 participants (4 males; age: 21.69 + 2.71) was collected in Experiment 1.”
- Line 218: “Another 21 participants (6 males; age: 22.24 + 2.16) ) were recruited in this experiment.”
Discussion:
Limits of the study should be discussed briefly (e.g., numbers of participants…).
Comments on the whole manuscript
- The authors should review the separation of words at the end of lines throughout the manuscript.
Lines 55-57: “Thus, the latent state maintenance, in combination with active state, likely manifested an evolution- ary advantage.”
ð evolutiona-ry
Lines 59-61: “The 59 working memory items can flexibly transition between the active and latent states accord-ing to the task contexts [12],[20-21].”
ð accor-ding
Lines 99-101: “We expected to replicate the dissociation account by observing no active load variation effect on latent mem-ories maintenance.”
ð memo-ries
Lines 126-127: “The memory response is rec-orded by pressing the keys.”
ð recor-ded
Lines 209-210: “That replicated previous study in a slightly modified retro-cue paradigm, conform- ing to dissociation account between active and latent states.”
ð confor-ming
Lines 270-271: “The results suggested that the vari-ation in active memory load affected latent memory performance, consistent with disso-ciation account.”
ð varia-tion
Lines 288-290: “The current study provided a new avenue for achieving perfect independ-ence of active and latent state, such that generating little memory loss in terms of overall memory performance.”
ð indepen-dence
Lines 291-293: “According to the current findings, there are some possible explanations for account-ing the influence of spatial proximity on the interaction between active and latent memory maintenance.”
ð accoun-ting
Lines 332-33: “Thus, the priority of memory items induced the separation of storage neural state follow-ing the presence of retro-cue during the memory information retention [40],[41].”
ð follo-wing
- Line 120: “Figure 1. the schematic of trial events in Experiment 1. The values below indicated the time span.”
- Lines 183-185: “Figure 2. behavioral accuracy in Experiment 1. The mean accuracy in the two memory arrays under the two load conditions (left). The mean accuracy in the active and latent states under the two load conditions. Error bars denote SEM.”
- Line 222: “Figure 3. the schematic of trial events in Experiment 2. The values below indicated the time span.”
- Lines 241-243: “Figure 4. behavioral accuracy in Experiment 2. The mean accuracy in the two memory arrays under the two load conditions (left). The mean accuracy in the active and latent states under the two load conditions. Error bars denote SEM. ."
The first word of the sentence must be capitalized. Moreover, the word "below" should be deleted, since the figures are located above the titles. The acronym SEM has to be explained. Finally, a period should be deleted at the end of the sentence on line 243.
ð Figure 1. The schematic of trial events in Experiment 1. The values indicated the time span.
ð Figure 3. The schematic of trial events in Experiment 2. The values indicated the time span.
ð Figure 4. Behavioral accuracy in Experiment 2. The mean accuracy in the two memory arrays under the two load conditions (left). The mean accuracy in the active and latent states under the two load conditions. Error bars denote SEM.
- The p-value should always be in italics, like all statistical values. A space is required before and after mathematical symbols (<, >, =). The authors should make the changes throughout the document.
Lines 187-191: “As plotted in Figure 2 (left), the results from a 2 (active load 3 vs. active load 4) × 2 (memory array 1 vs. memory array 2) repeated measures ANOVA showed a significant main effect of load, F (1, 22) = 21.281, p < 0.001, η2p= 0.492. The effect of memory type was also significant, F (1, 22) =9.135, p = 0.006, η2p = 0.492. But the interaction between load and memory type was not significant, F (1, 22) <1.
Lines 199-206: “A significant main effect of state was observed, F (1, 22) = 8.534, p = 0.008, η2p = 0.279, while the main effect of active load did not achieve significance, F (1, 22) = 4.440, p = 0.514, η2p = 0.020. Importantly, there was a significant interaction between the two factors, F (1, 22) = 5.870, p = 0.024, η2p = 0.211. A paired sample T-test analysis was then performed. The results suggested that the active memories suffer a great impairment as the active load had an increase, t(22) = 3.451, p = 0.002, Cohen’s d = 0.719.”
Lines 245-249: “First, the results from a 2 (active load 3 vs. active load 4) × 2 (memory array 1 vs. memory array 2) repeated measures ANOVA showed a significant main effect of load, F(1, 20) = 7.119, p = 0.015, η2p= 0.262. The effect of memory type was not significant, F (1, 20) =0.065, p = 0.802, η2p = 0.003. There also no interaction between load and memory type, F (1, 20) =0.030, p = 0.863, η2p = 0.002.”
- Lines 198-199: “What’s more, a 2 (active state vs. latent state) × 2 (active load 3 vs. active load 4) repeated measure ANOVA was conducted (Figure 2 right). “
The expression “What’s more” is used more in informal language. Authors should use “Moreover”.
- Line 218: Another 21 participants (6 males; age: 22.24 + 2.16) ) were recruited in this experiment.
It seems that the appropriate symbol is ± .
- Line 254: “we administrated a 2× 2 repeated measure ANOVA.”
The capital letter is missing at the beginning of the sentence.
ð “We administrated a 2× 2 repeated measure ANOVA.”
- Lines 254-257: “An interaction between the neural state and active load failed to reach significance, F (1, 20) = 0.821, p = 0.376, η2p =0.039. while we observe a significant main effect of neural state, F (1, 20) = 9.237, p = 0.006, η2p = 0.316, and active load, F (1, 20) = 10.764, p = 0.004, η2p = 0.350.”
The full stop before “while” should be deleted and replaced by a comma.
ð “An interaction between the neural state and active load failed to reach significance, F (1, 20) = 0.821, p = 0.376, η2p =0.039, while we observe a significant main effect of neural state, F (1, 20) = 9.237, p = 0.006, η2p = 0.316, and active load, F (1, 20) = 10.764, p = 0.004, η2p = 0.350.”
Citations in brackets have to be modified according instructions for authors:
For examples:
- Lines 52-55: “In terms of the energy consumption, persistent firing underlying the active memory retention is assumed to be energetically expensive, while short-term synaptic plasticity pattern of memory retention has a clear cost advantage with more efficiency [8],[12].”
ð [8, 12].
- Lines 57-59: “From the functional perspective, currently task-relevant information was held in the active state which was responsible for the immediate online processing, while latent state provided robust maintenance for prospective-relevant items [18],[19].”
ð [18, 19]
- Lines 59-61: “The 59 working memory items can flexibly transition between the active and latent states accord-ing to the task contexts [12],[20-21].”
ð [12, 20-21]
- Lines 332-33: “Thus, the priority of memory items induced the separation of storage neural state follow-ing the presence of retro-cue during the memory information retention [40],[41].”
ð [40,41]
- Lines 337-338: “The separation of neural storage state was based on the relevance of memory items to the immediate task [20],[23],[25].”
ð [20, 23, 25]
- Lines 368-370: “Institutional Review Board Statement: The study was conducted in accordance with the Declaration of Helsinki and approved by the Institutional Review Board (or Ethics Committee) of Liaoning Normal University (approval code: 20221221 ).”
The authors should delete the space before closing the parenthesis.
ð “Institutional Review Board Statement: The study was conducted in accordance with the Declaration of Helsinki and approved by the Institutional Review Board (or Ethics Committee) of Liaoning Normal University (approval code: 20221221).”
- Lines 371-372: “Informed Consent Statement: Informed consent was obtained from all subjects involved in the study.”
The section “Informed Consent Statement” should be written in bold.
ð “Informed Consent Statement: Informed consent was obtained from all subjects involved in the study.”
- Lines 198-199: “What’s more, a 2 (active state vs. latent state) × 2 (active load 3 vs. active load 4) repeated measure ANOVA was conducted (Figure 2 right). “
Authors use sometimes expression (e.g., “What’s more”...), used more in informal language. Authors should modify them. For this sentence, they should use “Moreover”.
Reviewer 2 Report
I have attached some specific, but minor comments about this manuscript. While this work is not a large leap forward in the field, I do feel that it moves the field forward in a meaningful, incremental way. Most of my comments were regarding minor grammatical errors, which largely didn't impact my ability to understand the manuscript, I just wanted to try and help the authors identify some minor changes that could be made to make their paper even more clear. In my opinion, this manuscript should be accepted with minor revisions as it provides a stepping stone for research on working memory and factors that impact the way in which our working memory functions (with respect to the state in which information is processed).

I touched on this above, but overall this paper is well-written. There were some grammatical errors, but only one or two really impacted the meaning of a sentence to a point that was confusing to me.
Reviewer 3 Report
see attached file

see attached file
